

# Morphology and molecular phylogeny of three new deep-sea species of *Chrysogorgia* (Cnidaria, Octocorallia) from seamounts in the tropical Western Pacific Ocean

Yu Xu[1,2,3,4], Zifeng Zhan[1,2,3] and Kuidong Xu[1,2,3,4]

[1] Laboratory of Marine Organism Taxonomy and Phylogeny, Institute of Oceanology, Chinese Academy of Sciences, Qingdao, China
[2] Laboratory for Marine Biology and Biotechnology, Pilot National Laboratory for Marine Science and Technology (Qingdao), Qingdao, China
[3] Center for Ocean Mega-Science, Chinese Academy of Sciences, Qingdao, China
[4] University of Chinese Academy of Sciences, Beijing, China

Corresponding author
Kuidong Xu, kxu@qdio.ac.cn

## ABSTRACT

Three new species of *Chrysogorgia* were discovered from seamounts in the tropical Western Pacific Ocean. *Chrysogorgia dendritica* sp. nov. and *Chrysogorgia fragilis* sp. nov. were collected from the Kocebu Guyot of the Magellan Seamount chain with the water depth of 1,821 m and 1,279–1,321 m, respectively, and *Chrysogorgia gracilis* sp. nov. was collected from a seamount adjacent to the Mariana Trench with the water depth of 298 m. They all belong to the *Chrysogorgia* "group A, Spiculosae" with rods distributed in body wall and tentacles, and differ from all congeners except *C. abludo Pante & Watling, 2012* by having a tree-shaped colony (vs. bottlebrush-shaped, planar or biflabellate). *Chrysogorgia dendritica* sp. nov. is unique in having a monopodial stem, the 1/3L branching sequence and the amoeba-shaped sclerites (sclerites branched toward to many directions) at the body bases of polyps. *Chrysogorgia fragilis* sp. nov. is most similar to *C. abludo*, but differs by the regular 1/3L branching sequence and elongate flat scales in coenenchyme. *Chrysogorgia gracilis* sp. nov. is easily separated from congeners by the 1/4L branching sequence, the absence of sclerites in the basal body wall, and the very sparse scales in coenenchyme. Based on the phylogenetic and genetic distance analyses of mtMutS gene, all the available *Chrysogorgia* species were separated into two main groups: one includes *C. binata*, *C.* cf. *stellata* and *C. chryseis*, which have two or more fans emerging from a short main stem (bi- or multi-flabellate colony); the other one includes all the species with the branching patterns as a single ascending spiral (clockwise or counterclockwise, bottlebrush-shaped colony), a fan (planar colony) and a bush of branches perched on top of a long straight stem (tree-shaped colony). Additionally, the tree-shaped colony represents a new branching pattern in *Chrysogorgia*, and therefore we extend the generic diagnosis.

## INTRODUCTION

The genus *Chrysogorgia Duchassaing & Michelotti, 1864* contains 72 species distributed in the world oceans, with water depths ranging from 10 m to 4,492 m (*Watling et al., 2011*; *Pante et al., 2012*; *Cairns, 2018*; *Xu et al., 2019*). Three branching forms have been recognized in the colonies of the genus: a single ascending spiral (clockwise or counterclockwise) producing a bottlebrush shape, a single fan (planar colony) and two fans emerging from a short main stem (biflabellate colony) (*Pante & Watling, 2012*; *Cordeiro, Castro & Pérez, 2015*). Based on the shapes of rods or scales in the body wall and tentacles, a rough grouping has been built for the separation of *Chrysogorgia* species. Versluys (*1902*) divided the genus *Chrysogorgia* into three groups, which were summarized by Cairns (*2001*) as "group A, Spiculosae", "group B, Squamosae aberrantes", and "group C, Squamosae typicae". *Cordeiro, Castro & Pérez (2015)* supplemented the fourth group "group D, Spiculosae aberrantes", which contains only the species *C. upsilonia Cordeiro, Castro & Pérez, 2015*.

While studying the benthic diversity in the tropical Western Pacific Ocean, we collected four specimens of *Chrysogorgia*. Based on morphological and phylogenetic analyses, we describe these specimens as three new species: *C. dendritica* sp. nov., *C. fragilis* sp. nov. and *C. gracilis* sp. nov. Their genetic distances and single mutations on mtMutS as well as phylogenetic relationships within *Chrysogorgia* species are discussed.

## MATERIALS & METHODS

### Specimen collection and morphological examination

Specimens were obtained by the remotely operated vehicle (ROV) *FaXian* (Discovery) from an unnamed seamount (temporarily named as M2) adjacent to the Mariana Trench and the Kocebu Guyot in the Magellan Seamounts in the tropical Western Pacific Ocean during the cruises of the R/V *KeXue* (Science) in 2016 and 2018 (Fig. 1). These specimens were photographed *in situ* before sampled, photographed on board and then stored in 75% ethanol after collection. Some branches were detached and stored at −80 °C for molecular analysis.

The general morphology and anatomy were examined by using a stereo dissecting microscope. The sclerites of the polyps and branches were isolated by digestion of the tissues in sodium hypochlorite, and then were washed with deionized water repeatedly. Polyps and sclerites were air-dried and mounted on carbon double adhesive tape and coated for the Scanning Electron Microscope (SEM) to investigate their structure. SEM scans were obtained and the optimum magnification was chosen for each kind of sclerites by using TM3030Plus SEM.

The morphological terminology follows *Bayer, Grasshoff & Verseveldt (1983)*, with which we coin the following new terms to describe the shape of sclerites. Tree-shaped colony: a bush of branches perched on top of a long straight stem, forming a tree shape. Example: *Chrysogorgia dendritica* sp. nov. (Fig. 2A). Amoeba-shaped sclerite: sclerites branched toward to many directions, varied in shape like an amoeba. Example: *Chrysogorgia dendritica* sp. nov. (Fig. 3C).

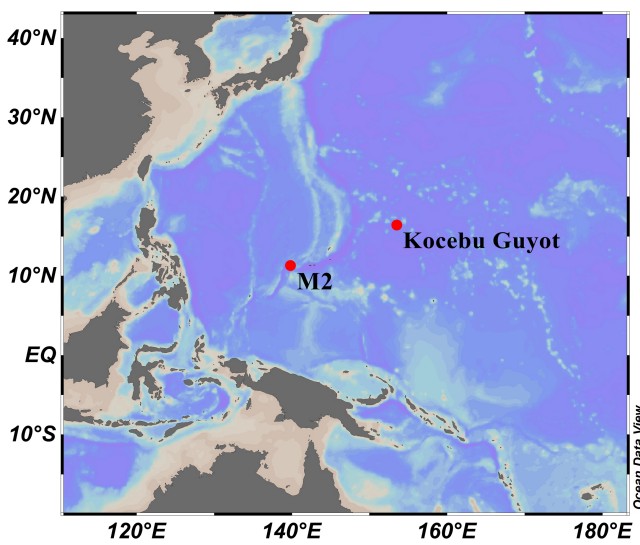

**Figure 1** **Sampling sites on a seamount (M2) adjacent to the Mariana Trench and the Kocebu Guyot in the Western Pacific Ocean.** (Map credit: ODV at http://odv.awi.de/, plotted by Yu Xu).

The type specimens of the three new species have been deposited in the Marine Biological Museum of Chinese Academy of Sciences (MBMCAS) at Qingdao, China.

## DNA extraction and sequencing

Total genomic DNA was extracted from the polyps of each specimen using the TIANamp Marine Animal DNA Kit (Tiangen Bio. Co., Beijing, China) following the manufacturer's instructions. PCR amplification for the mitochondrial genomic region 5′-end of the DNA mismatch repair protein − $mutS$ − homolog (mtMutS) was conducted using primers AnthoCorMSH (5′-AGGAGAATTATTCTAAGTATGG-3′; *Herrera, Baco & Sánchez, 2010*) and Mut-3458R (5′-TSGAGCAAAAGCCACTCC-3′; *Sánchez, Lasker & Taylor, 2003*). PCR reactions were performed using I-5$^{TM}$ 2× High-Fidelity Master Mix DNA polymerase (TsingKe Biotech, Beijing, China). The amplification cycle conditions were as follow: denaturation at 98 °C for 2 min, followed by 32 cycles of denaturation at 98 °C for 20 s, annealing at 50 °C for 20 s, extension at 72 °C for 15 s, and a final extension step at 72 °C for 2 min. PCR purification and sequencing were performed by TsingKe Biological Technology (TsingKe Biotech, Beijing, China).

## Genetic distance and phylogenetic analyses

The mtMutS may be the most variable mitochondrial gene in octocorals (*Herrera, Baco & Sánchez, 2010*; *McFadden et al., 2011*; *Li, Zhan & Xu, 2017*), and we selected this marker for molecular identification and phylogenetic analyses. All the available mtMutS sequences of *Chrysogorgia* spp. and the out-group species from related chrysogorgiid genera were downloaded from GenBank. The sequences from duplicate isolates or without associated publications or named *Chrysogorgia* sp. or containing sequencing errors (marked with ''n'' or ''y'' in the original sequences) were omitted from the molecular analyses. To correct

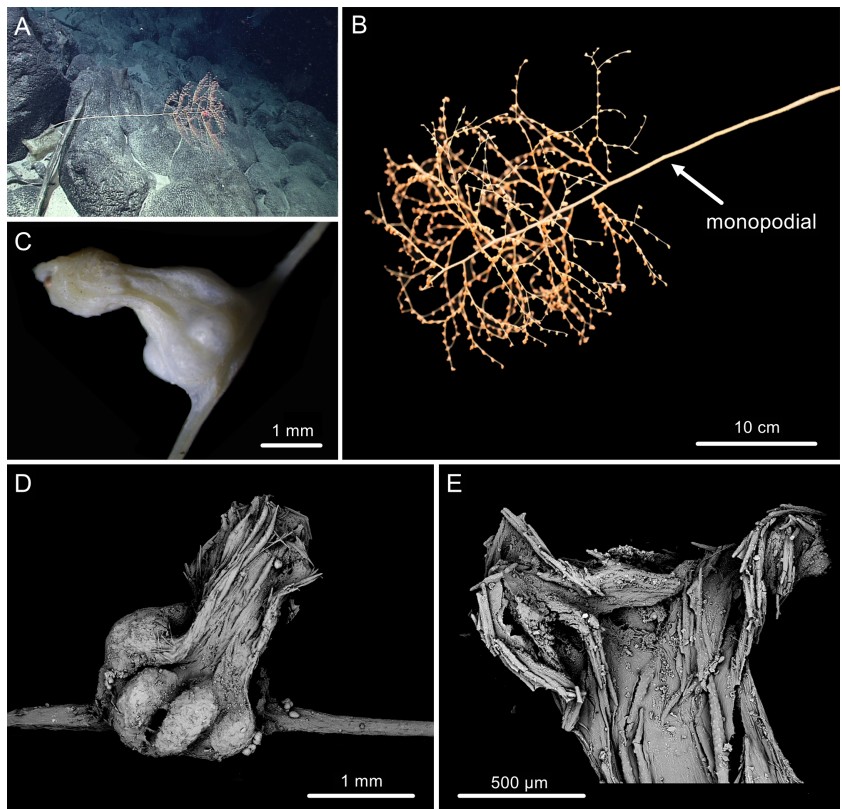

**Figure 2** **The external morphology and polyps of** *Chrysogorgia dendritica* **sp. nov.** (A) The holotype in situ; (B) The holotype immediately after collection; (C) A single polyp under light microscope; (D) Single polyp under SEM. (E) Tentacles with rods under SEM (Photo credit: Yu Xu and Shaoqing Wang).

possible mistakes, all the selected sequences were visually inspected, and translated to amino acids (AA) to insure all the AA sequences not including stop codons and suspicious substituteions. The nucleotide and AA sequences were aligned using MAFFT v.7 (*Katoh & Standley, 2013*) with the G-INS-i algorithm. With the guidance of the AA alignment, the nucleotide alignment was refined using BioEdit v7.0.5 (*Hall, 1999*), and only the nucleotide alignment was used in the subsequent analyses. Genetic distances, calculated as uncorrected "*p*" distances within each species and among species, were estimated using MEGA 6.0 (*Tamura et al., 2013*).

For the phylogenetic analyses, only one sequence was randomly selected from the conspecific sequences without genetic divergence. The evolutionary model GTR+G was the best-fitted model for mtMutS, selected by AIC as implemented in jModeltest2 (*Darriba et al., 2012*). Maximum likelihood (ML) analysis was carried out using PhyML-3.1 (*Guindon et al., 2010*). For the ML bootstraps, we consider values <70% as low, 70 −94% as moderate and ≥ 95% as high following *Hillis & Bull (1993)*. Node support came from a majority-rule consensus tree of 1,000 bootstrap replicates. Bayesian inference (BI) analysis was carried out using MrBayes v3.2.3 (*Ronquist & Huelsenbeck, 2003*) on CIPRES Science Gateway. Posterior probability was estimated using four chains running 10,000,000 generations

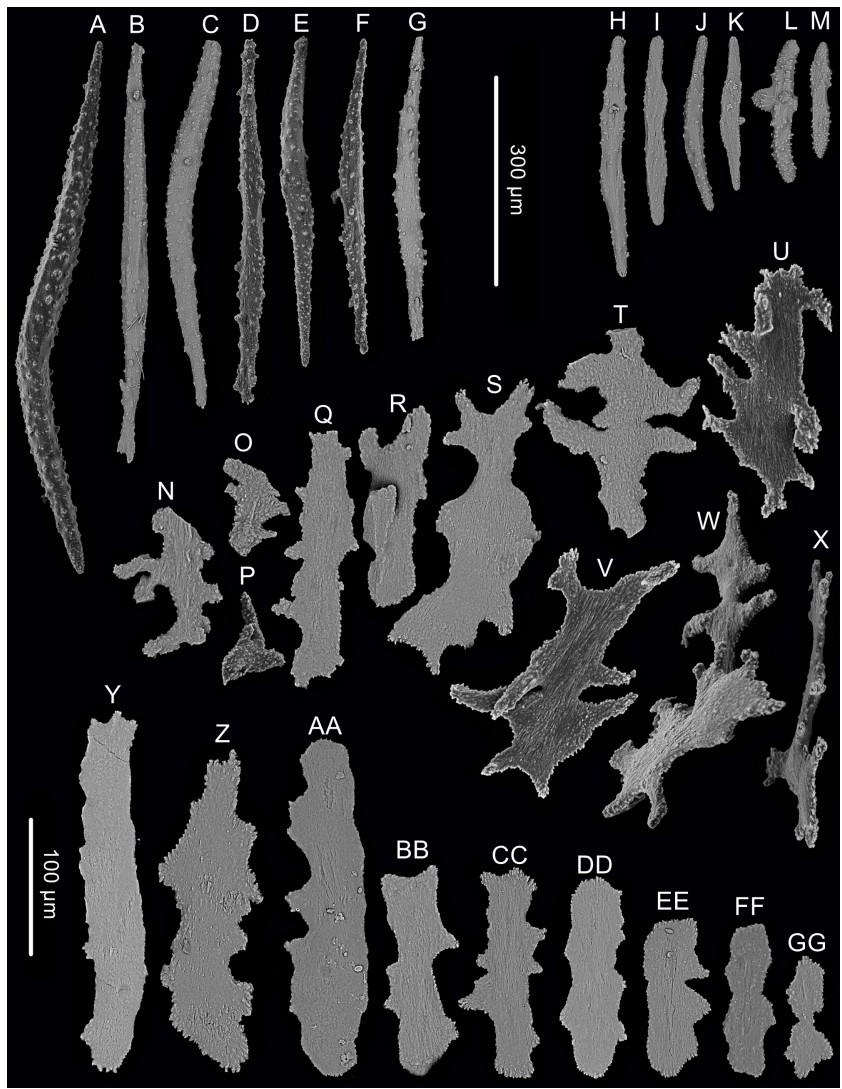

**Figure 3** **Sclerites of *Chrysogorgia dendritica*. sp. nov.** (A–G) Sclerites of the polyp neck. (H–M) Sclerites in the back of tentacles; (N–X) Sclerites at the body base; (Y–GG) Sclerites in coenenchyme. Scale bars: A–G and H–M, N–X and Y–GG at the same scale, respectively (Image credit: Yu Xu).

sampling every 1,000 generations. The first 25% of sampled trees were considered burn-in trees. Convergence was assessed by checking the standard deviation of partition frequencies (<0.01), the potential scale reduction factor (ca. 1.00), and the plots of log likelihood values (no obvious trend was observed over time). For the Bayesian posterior probabilities, we consider values <0.95 as low and ≥ 0.95 as high following *Alfaro, Zoller & Lutzoni (2003)*. The GenBank accession numbers of the mtMutS sequences were listed next to the species names in the phylogenetic tree.

## ZooBank registration

The electronic version of this article in Portable Document Format (PDF) will represent a published work according to the International Commission on Zoological Nomenclature (ICZN), hence the new names contained in the electronic version are effectively published under that Code from the electronic edition alone. This published work and the nomenclatural acts it contains have been registered in ZooBank, the online registration system for the ICZN. The ZooBank Life Science Identifiers (LSIDs) can be resolved and the associated information viewed through any standard web browser by appending the LSID to the prefix http://zoobank.org/. The LSID for this publication is: urn:lsid:zoobank.org:pub:00D5E053-EFF8-4142-8D16-AAFC17D028E2. The online version of this work is archived and available from the following digital repositories: PeerJ, PubMed Central, and CLOCKSS.

## RESULTS

**Class Anthozoa** *Ehrenberg (1834)*
**Subclass Octocorallia** *Haeckel (1866)*
**Order Alcyonacea** *Lamouroux (1812)*
**Suborder Calcaxonia** *Grasshoff (1999)*
**Family Chrysogorgiidae** *Verrill (1883)*
**Genus** *Chrysogorgia Duchassaing & Michelotti (1864)*

*Chrysogorgia dendritica* **sp. nov.** (Figs. 2 and 3; Table 1)
    urn:lsid:zoobank.org:act:F0050AD3-9E65-4B03-8D26-2C687018DCAD
**Holotype.** MBM286354, station FX-Dive 178 (17°20.18′N, 152°41.85′E), Kocebu Guyot, depth 1,821 m, 12 April 2018. GenBank accession number: MN510469.
**Diagnosis.** *Chrysogorgia* "group A, Spiculosae" with a long monopodial stem and a branching part on the top. Branching sequence 1/3L. Branches nearly perpendicular to stem, subdivided dichotomously. Polyps with a long neck and an expanded base. Rods and spindles in tentacles and polyp neck coarse with many warts. Scales and rare plates at polyp body base flat and amoeba-shaped. Scales in coenenchyme thin with irregular edges.
**Description.** Specimen about 57 cm long with the holdfast not recovered. Colony tree-shaped, composed of a 36 cm long, straight and unbranched stem and a 21 cm long branched part with branching sequence 1/3L. The whole stem monopodial from bottom to top with lateral branches producing on the top. Stem surface almost smooth with a strong golden metallic luster, about two mm in diameter at base (Figs. 2A, 2B). Branches subdivided dichotomously, up to seventh order, most broken after collection. Distance between adjacent branches 16–22 mm, and orthostiche interval 50–55 mm. First branch internodes 15–20 mm long, with the terminal branchlets up to 50 mm. Polyps with a long neck and an expanded base, about three mm long and two mm wide at bases, composed of one or two on the first internodes, one to five in middle internodes, and up to six in

Xu et al. (2020), *PeerJ*, DOI 10.7717/peerj.8832

**Table 1** Morphological comparisons between *C. averta*, *C. abludo*, *C. dendritica* sp. nov., *C. fragilis* sp. nov. and *C. gracilis* sp. nov.

| Characters/species | *C. averta* | *C. abludo* | *C. fragilis* sp. nov. | *C. dendritica* sp. nov. | *C. gracilis* sp. nov. |
|---|---|---|---|---|---|
| Group type | A | A | A | A | A |
| Colony shape | bottlebrush-shaped | bottlebrush-shaped or tree-shaped | tree-shaped | tree-shaped | tree-shaped |
| Branching sequence | 3/8L | 1/3, 1/4L, irregular | 1/3L | 1/3L | 1/4L |
| Interbranch distance (mm) | 9–13 | 4.3–15.0 | 15–22 | 16–22 | 2.0–4.5 |
| Orthostiche interval (mm) | 75–78 | No data | 50–65 | 50–55 | 11–16 |
| First branch internode (mm) | 13–19 | 6.1–16.0 | 15–22 | 15–20 | 3–7 |
| Polyps on internodes | 0–2 | 1–2 | 0–4 | 1–5 | 1–10 |
| Polyps on terminal branchlets | 1–3 | 1–6 | 1–10 | 1–6 | 3–20 |
| Polyps height (mm) | 1.1–1.9 | 0.8–2.2 | 2.0–4.0 | 3.0 | 0.9–1.5 |
| Sclerites in coenenchyme | rods and scales | small rugged scales | elongate flat scales | flat and lobed scales | elongated scales with smooth surface and edges |
| Sclerites in body wall | scales and rods | scales and rods | scales, rods and spindles | plates, scales, rods and spindles | scales and rods |
| Sclerites in tentacles | rods | rods | rods | rods | scales and rods |
| Distribution | North Atlantic | North Atlantic | Western Pacific | Western Pacific | Western Pacific |
| References | *Pante & Watling (2012)* | *Pante & Watling (2012)* | Present study | Present study | Present study |

terminal branchlets (Figs. 2C, 2D). No polyps on main axis internodes. Golden eggs often occurred at the expanded bases.

Rods longitudinally arranged in the back of tentacles, occasionally branched, with many small warts on surface, measuring 77–330 × 15–34 μm (Figs. 2E, 3H–3M). Sclerites rarely extend into the pinnules, which are otherwise sclerite-free. Spindles and rods longitudinally arranged in the long polyp neck, slender with many small warts on surface, usually slightly curved, measuring 193–800 × 25–56 μm (Figs. 3A–3G). Scales and rare plates transversely and crosswise arranged at body base, flat and amoeba-shaped with irregular edges, measuring 69–248 × 11–79 μm (Figs. 3N–Figs. 3X). Scales of coenenchyme sparse, flat and lobed with irregular edges, measuring 68–268 × 10–70 μm (Figs. 3Y–Figs. 3GG).

**Type locality.** Kocebu Guyot in the Magellan Seamount chain with water depth of 1,821 m.

**Etymology.** The Latin adjective *dendriticus* (dendritic) refers to the dendritic shape of the colony.

**Distribution and Habitat.** Found only on the Kocebu Guyot, where the colony attached to a died sponge (Fig. 2A). The water temperature was about 2.31 °C and the salinity about 35.8 psu.

**Remarks.** *Chrysogorgia dendritica* sp. nov. has a monopodial stem (Fig. 2B), which makes it appear to be a member of the chrysogorgiid genus *Metallogorgia Versluys, 1902*. However, the new species is characterized by a series of features matching the genus *Chrysogorgia Duchassaing & Michelotti, 1864*. These include the flexible branches, dichotomously subdivided branches not forming a sympodia, relatively large polyp with an expanded base and a narrow neck, and well differentiated coenenchyme usually with more sclerites. The new species also resembles to the genus *Pseudochrysogorgia Pante & France, 2010* in the monopodial stem, but differs by the obviously different polyps and the absence of ornamented sclerites. Our phylogenetic analysis (see below) supports this assignment as well. *Chrysogorgia dendritica* sp. nov. is distinctly different from congeners by its unique monopodial stem and the amoeba-shaped sclerites at the body bases.

### *Chrysogorgia fragilis* sp. nov. (Figs. 4 and 5; Table 1)
urn:lsid:zoobank.org:act:562CFDA7-88F5-4D81-8FE5-BDE1F56A3EEC

**Holotype.** MBM286351, station FX-Dive 172 (17° 23.64′N, 153°6.07′E), Kocebu Guyot, depth 1,321 m, 1 April 2018. GenBank accession number: MN510470.

**Paratype.** MBM286352, station FX-Dive 173 (17°28.12′N, 153° 10.07′E), Kocebu Guyot, depth 1,279 m, 7 April 2018.

**Diagnosis.** *Chrysogorgia* "group A, Spiculosae" with a long unbranched stem and a sympodial branching part with 1/3L branching sequence on the top. Branches subdivided dichotomously, up to fifth order. Polyps with an expanded base and a slender neck. Rods and spindles of the polyp neck and tentacles long and coarse, with many warts on surface. Scales at polyp body base elongated and thick, rarely branched. Scales in coenenchyme flat and elongated with irregular edges.

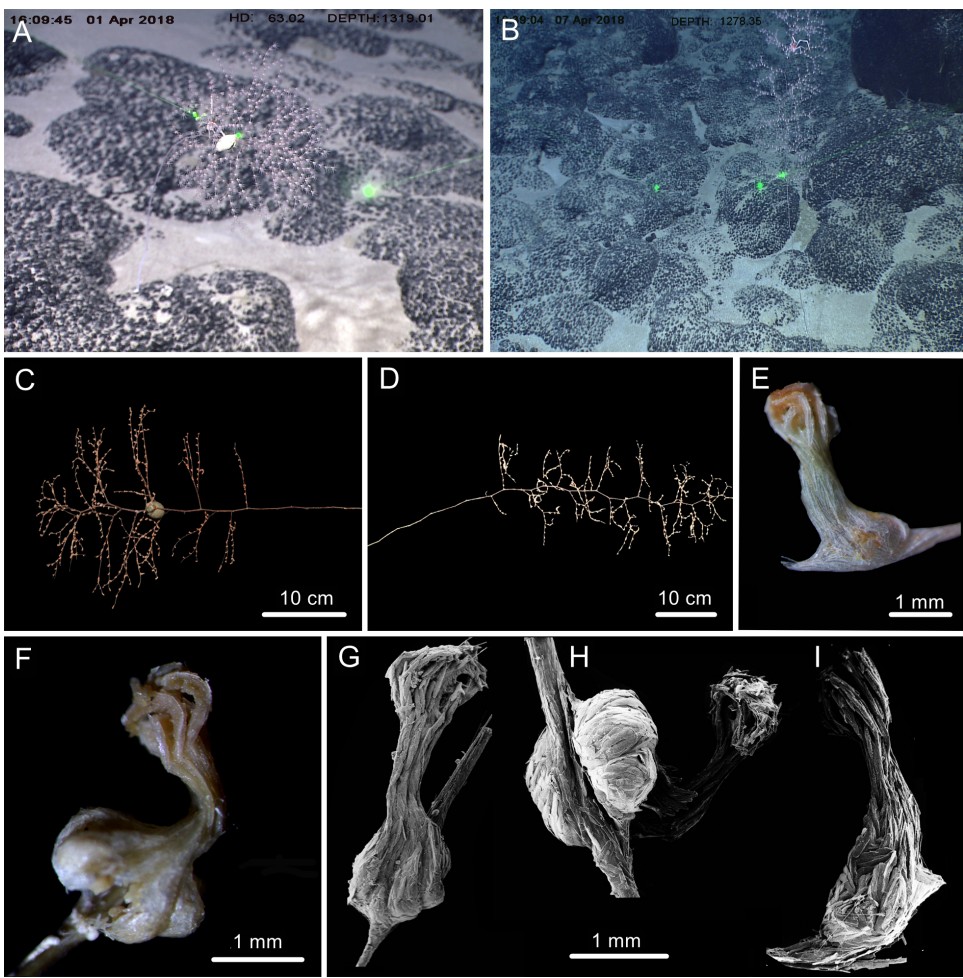

**Figure 4** **The external morphology and polyps of** *Chrysogorgia fragilis* **sp. nov.** (A) The holotype *in situ*. Laser dots spaced at 33 cm used for measuring dimensions; (B) The paratype *in situ*; (C) The holotype immediately after collection; (D) The paratype after fixation; (E, F) A single polyp under light microscope; (G–I) Three polyps under SEM (Photo credit: Yu Xu and Shaoqing Wang).

**Description.** Specimen of holotype about 55 cm in height excluding the holdfast. Colony tree-shaped, composed of a sympodial branching part on the top and a fragile, slender and unbranched stem about 35.5 cm long and 1.5 mm in diameter at base (Fig. 4C). Stem surface almost smooth with a few scars and aeruginous metallic luster, and sometimes covered with a layer of pink mucous membrane. Branching part produced a slightly zigzag pattern at the top portion with branching sequence 1/3L. Branches subdivided dichotomously, nearly perpendicular to the axis, up to fifth order, most broken after collection. Distance between adjacent branches and the first branch internodes both 15–22 mm long, orthostiche interval 50–65 mm, and the terminal branches up to 75 mm. Polyps with a long neck and an expanded body base, 2–4 mm long, 1–2 mm wide at base, with the neck up to two mm long and less than one mm wide (Figs. 4E–Figs. 4I). Up to two polyps on the first internodes, two to four in middle internodes, up to ten in terminal branchlets.

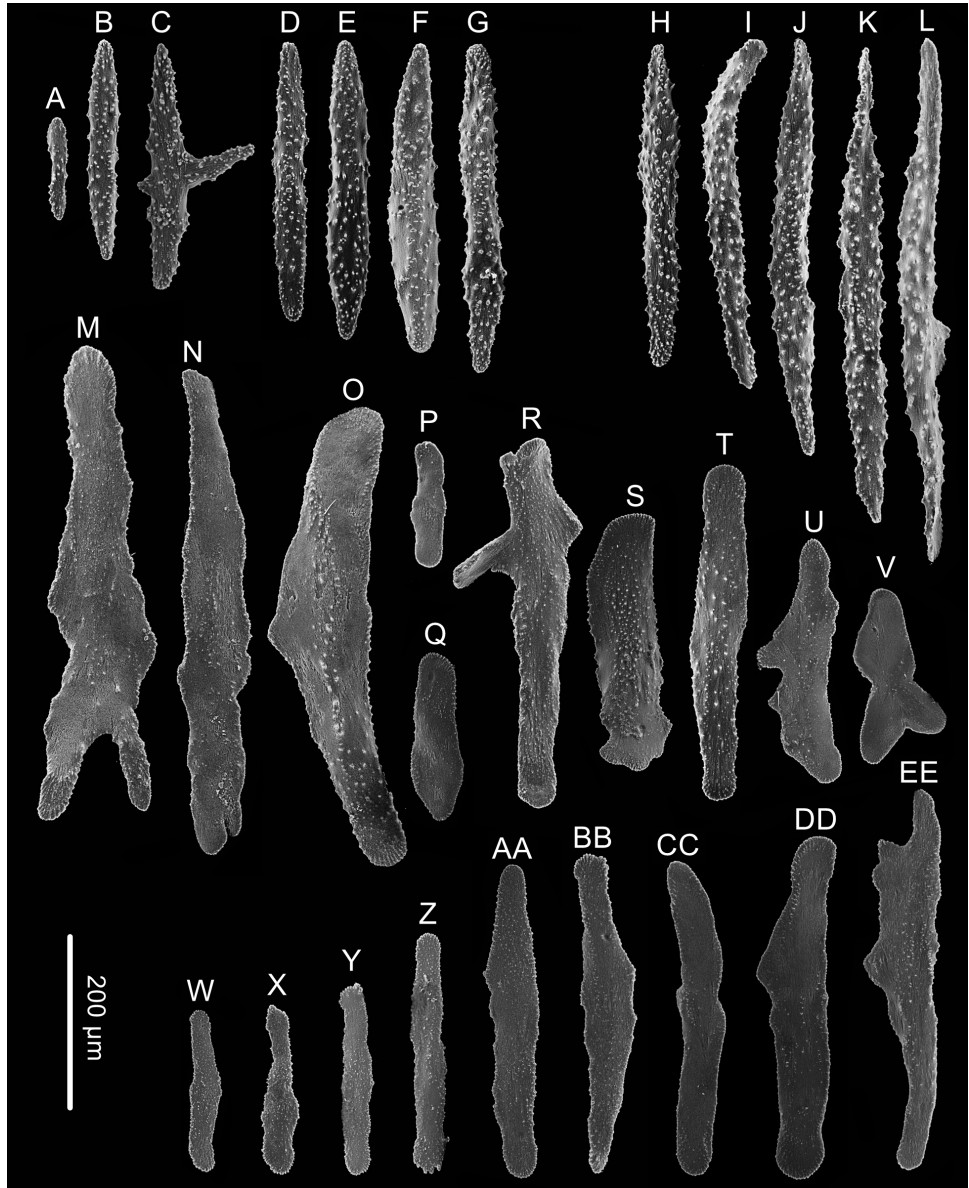

**Figure 5** **Sclerites of *Chrysogorgia fragilis* sp.nov.** (A–G) Sclerites of the polyp neck; (H–L) Sclerites in the back of tentacles; (M–V) Sclerites at the expanded polyp body base; (W–EE) Sclerites in coenenchyme. Scale bar: all at the same scale (Image credit: Yu Xu).

No polyp on main axis internodes. Golden eggs present in expanded body bases. Polyps pink immediately after collection, color gradually faded in alcohol.

Rods longitudinally arranged in the back of the tentacles, rarely branched, with many small warts on surface, measuring 105–442 × 14–50 μm (Figs. 4G, 4I, 5A–5G). Rare sclerites extend into the pinnules, and pinnules free of sclerites. Spindles and rods longitudinally arranged in the polyp neck, slender with many small warts on surface, sometimes with one or two sharp ends, measuring 170–600 × 17–62 μm (Figs. 4H,

5H–5L). Scales longitudinally and transversally arranged at base of expanded polyp body, elongated with a few warts and irregular edges, sometimes branched, thicker and wider than those in coenenchyme, measuring 144–551 × 34–106 μm (Figs. 4H, 5M–5V). Scales of coenenchyme flat and elongate, rarely with distinctly irregular edges, measuring 122–435 × 28–83 μm (Figs. 5W–5EE).

**Variation of Paratype.** Specimen 65 cm in height with unbranched stem about 35 cm long and one mm across at base (Fig. 4D). Branching part relatively longer and more zigzagging.

**Type locality.** Kocebu Guyot in the Magellan Seamount chain with water depths of 1,279–1,321 m.

**Etymology.** The Latin adjective *fragilis* (fragile) refers to the fragile stem and branches of the species.

**Distribution and habitat.** Found only on the Kocebu Guyot in the Magellan Seamount chain. Colonies attached to rocky substrate. The holotype was attached with an egg-shaped structure and the paratype with an individual of the crustacean genus *Galathea* Fabricius, 1793 (Figs. 4A, 4B). The water temperature was about 3.2 °C and the salinity about 35.8.

**Remarks.** *Chrysogorgia fragilis* sp. nov. belongs to the "group A, Spiculosae" with an unusual branching sequence of 1/3L, with which it is similar to *Chrysogorgia midas Cairns, 2018* and *C. dendritica* sp. nov. However, the new species differs distinctly from *C. midas Cairns, 2018* by the tree-shaped colony (vs. bottlebrush-shaped), wider orthostiche interval (50–65 mm vs. 12–18 mm), larger polyps (2.0–4.0 mm vs. 1.1 mm), and the presence of various shapes of scales at the body bases (vs. absence). *Chrysogorgia fragilis* sp. nov. is also similar to *C. abludo Pante & Watling, 2012* and *C. averta Pante & Watling, 2012*, two species found in the north-western Atlantic Ocean, in possessing the wide orthostiche interval and long and straight unbranched stem. However, the new species is easily separated from *C. averta* by the tree-shaped colony (vs. bottlebrush-shaped). It differs from *C. abludo* by the regular 1/3L branching sequence (vs. irregular) and elongate flat scales in coenenchyme (vs. small rugged scales) (Table 1). *Chrysogorgia fragilis* sp. nov. differs from *C. dendritica* sp. nov. by a sympodial branching part (vs. monopodial) and relatively regular scales at the body bases (vs. amoeba-shaped).

*Chrysogorgia gracilis* **sp. nov.** (Figs. 6 and 7; Table 1)
urn:lsid:zoobank.org:act:F557CE43-D43C-4E5F-86C1-3EFE330A9443

**Holotype:** MBM286350, station FX-Dive 57 (11°18.34′N, 139°21.43′E), an unnamed seamount (temporarily named as M2) adjacent to the Mariana Trench, depth 298 m, 23 March 2016. GenBank accession number: MN510472.

**Diagnosis:** *Chrysogorgia* "group A, Spiculosae" with a long unbranched stem and a sympodial branching part emanating in a regular 1/4L spiral on the top. Stem and branches slender, with branches subdivided dichotomously. Terminal branchlets gracile and somewhat whip-like. Polyps small and thin, no more than 1.5 mm long, located on one side of branches. Rods and rod-like scales slender and abundant in tentacles and at the bases of tentacles. No sclerites at polyp body base. Scales elongated, rare to absent

Peer

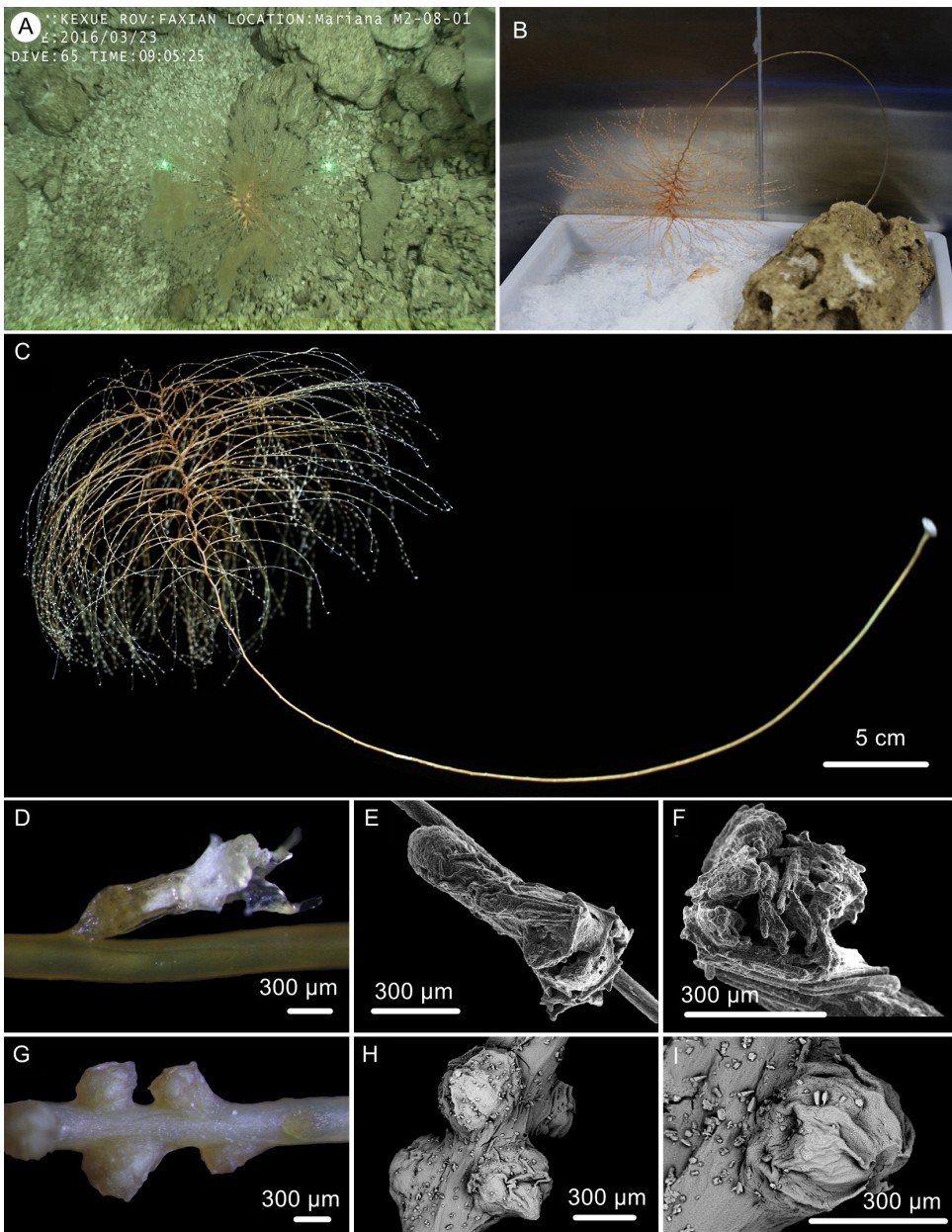

**Figure 6** **The external morphology and polyps of the holotype of *Chrysogorgia gracilis* sp. nov.** (A–C) The holotype in situ (A) and after collection (B) and fixation (C); Laser dots spaced at 33 cm used for measuring dimensions; (D) A single polyp under light microscope; (E) A single polyp under SEM; (F) Tentacles under SEM; (G) Mesozooids at the base of branch under light microscope; (H) Four mesozooids under SEM; (I) A single mesozooid under SEM (Photo credit: Yu Xu and Shaoqing Wang).

in coenenchyme. Mesozooids dense along the internodes of top stem and the bases of branches.

**Description.** Specimen orange to reddish after collection, became yellow in alcohol, about 51.8 cm long (Figs. 6B, 6C). Stem and branches golden with slightly glaucous metallic

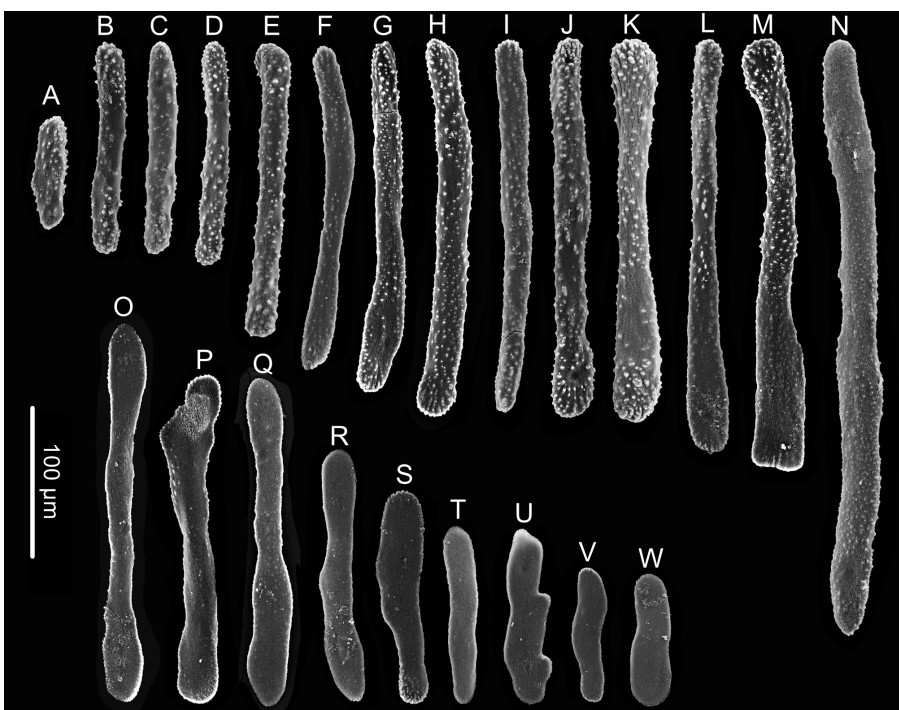

**Figure 7** **Sclerites of *Chrysogorgia gracilis* sp. nov.** (A–N) Sclerites in tentacles and at the bases of tentacles; (O–W) Sclerites in coenenchyme. Scales bars: all at the same scale (Image credit: Yu Xu).

luster. Colony tree-shaped. Unbranched stem curved, up to 40.5 cm in arc length and 1.0–2.9 mm in diameter, emanating in regular 1/4L spiral on the top of a tall (Fig. 6C). Holdfast small and rounded, about 9.8–12.5 mm in diameter. Distance between adjacent branches in stem 2.0–4.5 mm long and orthostiche interval 11–16 mm. The first branch internodes 3–7 mm. Branches subdivided 2–7 times and the angle between bifurcating branches particularly obtuse: 18°–62° . Terminal branchlets slender, usually whip-like, up to 90 mm long.

Polyps translucent, 0.9–1.5 mm long, 0.2–0.4 mm wide, uniserial spaced 2–5 mm on the branches by one side, with angle random to the branches. Polyp body base golden, without sclerites (Fig. 6D). Tentacles up to 1.0 mm in length, became white in alcohol. Three to 20 polyps on terminal branchlets and up to ten polyps in branch internodes. Axial internodal polyps not observed in the stem, where dense mesozooids occurred along the internodes of the stem and branch bases. Mesozooids bud-like shaped, orange in situ and yellowish in alcohol, without sclerites, about 0.3–0.5 mm wide and up to 0.4 mm high (Figs. 6A, 6G–6I).

Rods and rod-like scales slender, sometimes one end flat and the other end cylindric, mostly aggregated in the joints between the tentacles and bodies, or longitudinally along the back of the tentacles, with dentate projections at one or both ends and coarse, granular warts on surface, measuring 90–450 × 15–20 µm (Figs. 6E, 6F, 7A–7N). Coenenchyme in branches with a thin pellucid and calcareous layer in outside of the central axis, sometimes

with regular scales oriented along branches or without scales on branches. Scales elongated with smooth surface and edges, occasionally with finely serrated edges, usually becoming narrow in middle, rare to absent in coenenchyme, measuring $50–250 \times 12–38$ μm (Figs. 7O–7W). All sclerites colorless.

**Type locality.** An unnamed seamount (temporarily named as M2) adjacent to the Mariana Trench with water depths of 298 m.

**Etymology.** The Latin adjective *gracilis* (gracile) refers to the gracile stem and branches of this species.

**Distribution and Habitat.** Found only on the M2 seamount adjacent to the Mariana Trench. Colony attached to a rocky substrate with a small holdfast (Fig. 6A).

**Remarks.** Among the species possessing 1/4L branching sequence and rods in tentacles, *C. gracilis* sp. nov. mostly resembles *C. pyramidalis Kükenthal, 1908* in the same branching division and similar length, soft and translucent polyp's body, and the very rare sclerites in coenenchyme (*Kükenthal, 1908*; *Kinoshita, 1913*; *Cairns, 2001*). However, *C. gracilis* sp. nov. differs from *C. pyramidalis* by its distinctly longer and unbranched stem, more slender rods with lobed or irregular round ends, nearly smooth and elongated scales in coenenchyme, and the presence of mesozooids (*Kükenthal, 1908*; *Kinoshita, 1913*). Compared with congeners, *Chrysogorgia gracilis* sp. nov. possesses much thinner and smaller polyps, where no sclerites occur at the bases, and rare to absent sclerites in coenenchyme. In contrast, both the polyp body wall and coenenchyme are usually composed of numerous sclerites in other species of *Chrysogorgia*.

The specimen collected is characteristic in having numerous yellowish mesozooids on the stem internodes and the bases of branches (Figs. 6G, 6I). The mesozooids in this species are distinguished from the nematozooids or cnidae existed in some species of *Chrysogorgia* and *Iridogorgia Verrill, 1883* in size, shape and distribution (*Kinoshita, 1913*; *Deichmann, 1936*). The nematozooids are a kind of small protuberances or verrucae distributed on the surface of polyps and coenenchyme on branches, while the mesozooids are similar to polyps in width and are independent on the surface of branches (Fig. 6A).

## Genetic distance and phylogenetic analyses

The mtMutS sequences of the three new species were obtained and deposited in GenBank, with the accession number and the length are as follows: MN510469, 620 bp for *Chrysogorgia dendritica* sp. nov.; MN510470, 635 bp for *C. fragilis* sp. nov.; and MN510472, 635 bp for *C. gracilis* sp. nov. The alignment dataset comprised 623 nucleotide positions. The present intraspecific distances were calculated based on *C. abludo*, *C. tricaulis*, *C. artospira*, *C. averta* and *C. chryseis* populations, and no intraspecific variability was observed for the four species (Table 2). The mtMutS genetic distances among the species of *Chrysogorgia* range from zero to 2.42% (Table 2). The genetic distances between the new species *C. fragilis* sp. nov. and congeners are in the range of 0.16%–2.26%, and those between *C. gracilis* sp. nov. and congeners are in the range of 0.48%–2.10% (Table 2). No genetic variability was observed between *dendritica* sp. nov. and *C. abludo*, and the genetic distances between this new species and the rest congeners range from 0.16% to 2.42% (Table 2).

Xu et al. (2020), *PeerJ*, DOI 10.7717/peerj.8832

**Table 2  Interspecific and intraspecific uncorrected pairwise distances at mtMutS of species of *Chrysogorgia*.**

| | | 1 | 2 | 3 | 4 | 5 | 6 | 7 | 8 | 9 | 10 | 11 | 12 | 13 |
|---|---|---|---|---|---|---|---|---|---|---|---|---|---|---|
| 1 | ***Chrysogorgia gracilis* sp. nov.** MN510472 | – | | | | | | | | | | | | |
| 2 | ***C. dendritica* sp. nov.** MN510469 | 0.97% | – | | | | | | | | | | | |
| 3 | ***C. fragilis* sp. nov.** MN510470 | 1.13% | 0.16% | – | | | | | | | | | | |
| 4 | *C. abludo* GQ180139, JN227999 | 0.97% | 0 | 0.16% | 0 | | | | | | | | | |
| 5 | *C. tricaulis* JN227998, JN227990, JN227991, GQ180123–GQ180131, EU268056 | 0.65% | 0.97% | 0.81% | 0.97% | 0 | | | | | | | | |
| 6 | *C. artospira* GQ180132– GQ180135, GQ353317 | 0.48% | 0.81% | 0.65% | 0.81% | 0.16% | 0 | | | | | | | |
| 7 | *C. pinnata* JN227988 | 0.48% | 0.81% | 0.65% | 0.81% | 0.16% | 0 | – | | | | | | |
| 8 | *C. averta* KC788265, GQ180136 | 0.81% | 1.13% | 0.97% | 1.13% | 0.48% | 0.32% | 0.32% | 0 | | | | | |
| 9 | *C. ramificans* MK431863 | 1.13% | 1.45% | 1.29% | 1.45% | 0.81% | 0.65% | 0.65% | 0.97% | – | | | | |
| 10 | *C. monticola* JN227989 | 1.13% | 1.45% | 1.29% | 1.45% | 0.81% | 0.65% | 0.65% | 0.97% | 0.32% | – | | | |
| 11 | *C. binata* MK431862 | 2.10% | 2.42% | 2.26% | 2.42% | 1.77% | 1.61% | 1.61% | 1.94% | 2.26% | 2.26% | – | | |
| 12 | *C. cf. stellata* JN227920 | 1.94% | 2.26% | 2.10% | 2.26% | 1.61% | 1.45% | 1.45% | 1.77% | 2.10% | 2.10% | 0.16% | – | |
| 13 | *C. chryseis* JN227992, DQ297421 | 2.10% | 2.42% | 2.26% | 2.42% | 1.77% | 1.61% | 1.61% | 1.94% | 2.26% | 2.26% | 0.48% | 0.32% | 0 |

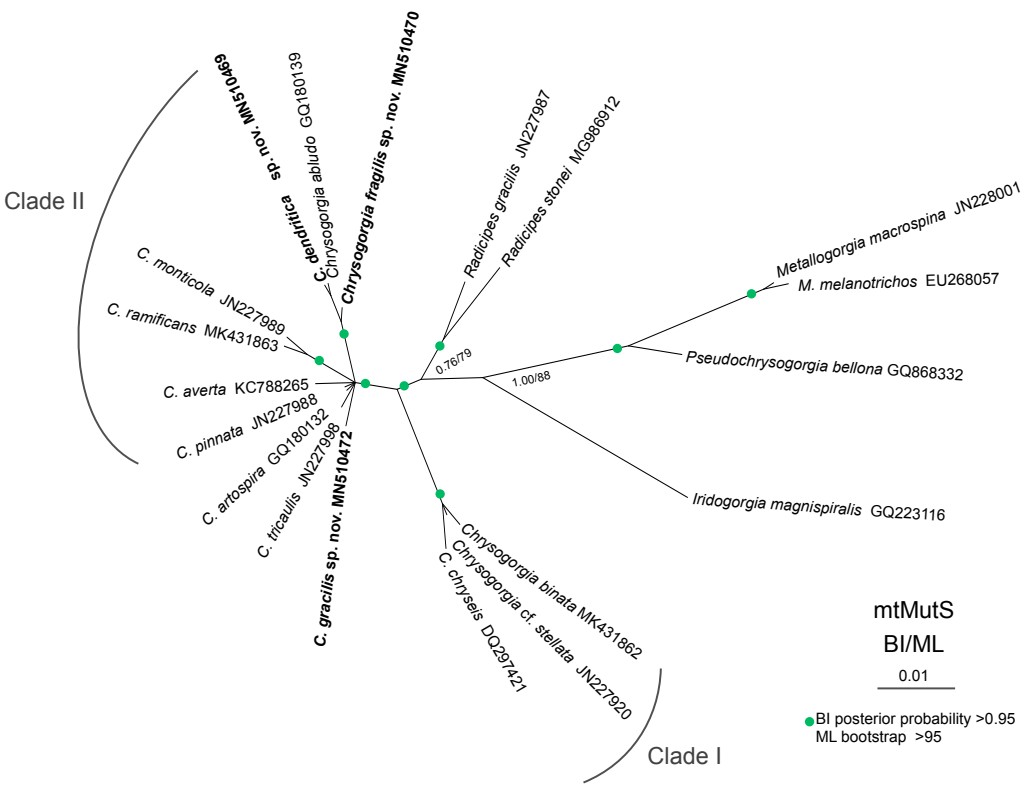

**Figure 8** **Maximum likelihood (ML) tree inferred from the mtMutS sequences of *Chrysogorgia* and the related species sequences.** The Bayesian inference (BI) tree and the ML tree are identical in topology. Node support is as follows: BI posterior probability/ML bootstrap. Newly sequenced species are in bold (Image credit: Zifeng Zhan).

The ML and BI phylogenetic trees are identical in topology, and thus only the former with the both support values was showed (Fig. 8). The *Chrysogorgia* species were separated into two main clades (Clade I and II) with high support values. Clade I includes *C. binata*, *C.* cf. *stellata* and *C. chryseis*, and Clade II contains all the rest species. The new species *C. dendritica* sp. nov. and *C. abludo* formed a sister subclade, followed by *C. fragilis* sp. nov. *Chrysogorgia gracilis* sp. nov. formed a sister subclade with *C. tricaulis*, *C. artospira*, *C. pinnata*, *C. averta* and the subclade *C. ramificans* + *C. monticola*.

## DISCUSSION

Both the morphology and molecular phylogenetic analysis supported the assignment of the three new species to the genus *Chrysogorgia*. The genetic distance analysis of mtMutS is considered as one of the first steps in an integrative identification of octocorals (*McFadden et al., 2011*; *Pante et al., 2012*). In the present study, however, the mtMutS genetic distances within *Chrysogorgia* are relatively low, and there is no barcoding gap (intraspecific zero vs. interspecific 0–2.42%) for species separation. Alternatively, single mutations on mtMutS, corresponding to the genetic distance of ca. 0.16%, can be used to separate *Chrysogorgia* species (*Pante & Watling, 2012*; *Pante et al., 2015*; this study). *Chrysogorgia gracilis* sp. nov.

and *C. fragilis* sp. nov. showed at least one single mutation difference from congeners (the corresponding genetic distances in range of 0.16%–2.26%; Table 2), supporting the establishment of the two new species. Although no genetic variability was observed between *C. dendritica* sp. nov. and *C. abludo*, the former is distinctly different from the latter by its unique monopodial stem and the amoeba-shaped sclerites at the polyp body bases.

Based on the diagnosis sensu *Pante & Watling, 2012* and *Cordeiro, Castro & Pérez, 2015*, the genus *Chrysogorgia* includes three branching forms: a single ascending spiral (clockwise or counterclockwise), a fan (planar colony) and two fans emerging from a short main stem (biflabellate colony). Based on the phylogenetic analysis, all the available *Chrysogorgia* species could be separated into two groups (Clade I and II). All species in Clade I (*C. binata*, *C.* cf. *stellata* and *C. chryseis*) have a bi- or multi-flabellate colony, as in the type species *C. desbonni* *Duchassaing & Michelotti, 1864*. The other species of *Chrysogorgia* possessing either a bottlebrush-shaped, a planar or a tree-shaped colony formed the Clade II with high support (Fig. 8). Furthermore, the genetic distances between Clade I and II are much higher than the intra-clade ones of Clade I (1.45%–2.42% vs. 0–0.48%; Table 2). Likely, Clade II represents a new subgroup of *Chrysogorgia* or even a new genus. However, only the sequences from 12 of 75 *Chrysogorgia* species are available for the genetic analysis. Further integrated genetic and morphological analyses are needed to verify this suggestion.

It is worth of note that all the new species possess a tree-shaped colony (monopodial, sympodial), which represent a new colony form in *Chrysogorgia*. Such a colony occurs also in the paratype NAS204-1 of *C. abludo* (*Pante & Watling, 2012*). Therefore, we add the tree-shaped colony to the diagnosis of the genus. Here, we extend the diagnosis of *Chrysogorgia* on the basis of *Pante & Watling (2012)* and *Cordeiro, Castro & Pérez (2015)*: Colony branching usually sympodial, occasionally monopodial, arising from a single ascending spiral (clockwise or counterclockwise, bottlebrush-shaped colony), a fan (planar colony), two fans emerging from a short main stem (biflabellate colony), or an unbranched main stem forming a tree-shaped colony. Axis with a metallic shine, dark to golden in color. Branch subdivided dichotomously or pinnately. Most polyps relatively large to the size of the branches they sit on, few in number and well separated from one another. Sclerites in the form of spindles, rods, scales and rare plates with little ornamentation.

## CONCLUSIONS

Based on the morphological and phylogenetic analyses, the newly sampled specimens are recognized as three new species *Chrysogorgia dendritica* sp. nov., *C. fragilis* sp. nov. and *C. gracilis* sp. nov. Furthermore, the tree-shaped colony of the new species represents a new branching pattern of *Chrysogorgia*, and therefore we extend the generic diagnosis.

## ACKNOWLEDGEMENTS

We thank the assistance of the crew of R/V *KeXue* and ROV *FaXian* for sample collection. We also thank Dr. Yang Li for comments on an early manuscript and Mr. Shaoqing Wang for taking the photos on board.

### Funding

This work was supported by the Key Program of National Natural Science Foundation of China (No. 41930533), the Strategic Priority Research Program of the Chinese Academy of Sciences (XDA19060401), the Science & Technology Basic Resources Investigation Program of China (2017FY100804) and the Senior User Project of RV KEXUE. The funders had no role in study design, data collection and analysis, decision to publish, or preparation of the manuscript.

### Grant Disclosures

The following grant information was disclosed by the authors:
Key Program of National Natural Science Foundation of China: 41930533.
Strategic Priority Research Program of the Chinese Academy of Sciences: XDA19060401.
Science & Technology Basic Resources Investigation Program of China: 2017FY100804.
RV KEXUE.

### Competing Interests

The authors declare there are no competing interests.

### Author Contributions

- Yu Xu conceived and designed the experiments, performed the experiments, analyzed the data, prepared figures and/or tables, authored or reviewed drafts of the paper, and approved the final draft.
- Zifeng Zhan analyzed the data, prepared figures and/or tables, authored or reviewed drafts of the paper, and approved the final draft.
- Kuidong Xu conceived and designed the experiments, authored or reviewed drafts of the paper, and approved the final draft.

### Data Availability

The DNA sequence of the four specimens and additional data is available in the Supplementary Files.

The specimens described in this study are available at the Marine Biological Museum of Chinese Academy of Sciences (MBMCAS) at Institute of Oceanology, Qingdao, China. Voucher IDs:

Chrysogorgia dendritica: MBM286354; *Chrysogorgia fragilis* holotype: MBM286351; *Chrysogorgia fragilis* paratypes: MBM286352; *Chrysogorgia gracilis*: MBM286350.

The mtMuts sequences of the new species are available at NCBI GenBank: MN510469, MN510470, and MN510472.

### New Species Registration

The following information was supplied regarding the registration of a newly described species:

Publication LSID: urn:lsid:zoobank.org:pub:00D5E053-EFF8-4142-8D16-AAFC17D028E2.

*Chrysogorgia dendritica* sp. nov. LSID: urn:lsid:zoobank.org:act:F0050AD3-9E65-4B03-8D26-2C687018DCAD

*Chrysogorgia fragilis* sp. nov. LSID: urn:lsid:zoobank.org:act:562CFDA7-88F5-4D81-8FE5-BDE1F56A3EEC

*Chrysogorgia gracilis* sp. nov. LSID: urn:lsid:zoobank.org:act:F557CE43-D43C-4E5F-86C1-3EFE330A9443

## Supplemental Information

Supplemental information for this article can be found online at http://dx.doi.org/10.7717/peerj.8832#supplemental-information.

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
