# Peer review of "Morphology and molecular phylogeny of three new deep-sea species of Chrysogorgia (Cnidaria, Octocorallia) from seamounts in the tropical Western Pacific Ocean"

_PeerJ, doi:10.7717/peerj.8832_

## Round 0.1 · original submission · Minor Revisions

I have heard back from two reviewers, both of whom have offered constructive comments on your work. As well, please check the annotated PDF from reviewer 1, and ensure you respond in detail to all comments with details in the responses file upon next submission. I look forward to seeing your revised paper.

Reviewer 1 ·

Basic reporting

.

Experimental design

.

Validity of the findings

.

Additional comments

Morphology and molecular phylogeny of three new deep-sea species of Chrysogorgia (Cnidaria, Octocorallia) from seamounts in the tropical Western Pacific Ocean
Yu Xu1,2,3,4, Zifeng Zhan1,2,3, Kuidong Xu1,2,3,4

Reviewer Comment
1. This is an excellent paper that contributes significantly to our taxonomic knowledge of the Octocorallia, and offers descriptions of three new species of the genus Chrysogorgia.

2. They have created some terms, but unfortunately have not defined them (i.e. amoeba-shaped sclerites, skateboard shaped scales, tree-shaped colony).

I think that when creating a new term, it is necessary to examine the terms of the previous authors and then define the term properly.

3. For octocoral taxonomy, sclerites need to write down the shapes, sizes, amounts, and arrangements for each of the following portions.

Polyp body, tentacle, pinnule, possibly the basal portion of the polyp, surface layer of the axis or branches (e.g. Pinnules sclerites can be species characters, but are not included in this manuscript.).

It is necessary to write them properly, or if there is a series of common bones, it is necessary to accurately describe the state.

Sclerites pictures must also be shown for each portion.
(There is concern about “Sclerites of the polyp neck and tentacles” in Fig. 4A, 6A.
In this case, it is necessary to distinguish between “polyp neck” and “tentacle, and information on the “poly body” is required).

4. Chrysogorgia is a genus of “sympodial branching”.

In contrast, Pseudochrysogorgia Pante & France, 2010, has been established for species that have the characteristics of Chrusogorgia and are monopodial branching.

While they write that C. dendritica is monopodial, and they cited Pante & France, 2010, moreover they introduced Pseudochrysogorgia in Fig. 9, but there is no textual explanation.

Need the explanation that C. dendritica is Chrisogorgia, not Pseudochrysogorgia.

I also suggest that the picture which can confirm that C. dendritica is monopodial is needed.

5. The References section needs some modification.

6. I suggest that they refer to the comments I wrote in the manuscript, and that they will be modified.


I hereby recommend this paper for publication in the Peer J.

Annotated reviews are not available for download in order to protect the identity of reviewers who chose to remain anonymous.

·

Basic reporting

no comment

Experimental design

no comments

Validity of the findings

no comment

Additional comments

The description of three new species from deep waters is an important contribution to the biodiversity research and make clear the relevance of these deep water environments that should be preserved. Every new exploration gives new species to science. The authors follow the taxonomic standards for the identification and description of the new species with an integrative approach. I just have two comment about the figures:
1-Personally, I like to see the size of the scales on the plates, not in the caption of the figure.
2-About the SEMs of polyps and branches, images like the ones in Fig. 3, do not show much detail, the structures do not look clean and they are collapsed. I suggest to fix the fragments for SEM in formalin for some hours, wash, and follow a dehydration protocol for critical point drying. I think that you will obtain very nice pictures in the future.

---

## Round 0.2 · accepted · Accept

The manuscript has been well revised, and I am happy to move this work into production. Thanks again for your nice submission.